# Post-Calibration Uncertainty Analysis for Travel Times at a Naval Weapons Industrial Reserve Plant

**Bulbul Ahmmed** [1,2] ⬤**, Scott C. James** [1,3,*] ⬤ **and Joe Yelderman** [1]

[1] Department of Geosciences, Baylor University, Waco, TX 76706, USA; ahmmedb@lanl.gov (B.A.); Joe_Yelderman@baylor.edu (J.Y.)

[2] Computational Earth Science Group, Los Alamos National Laboratory, Los Alamos, NM 87545, USA

[3] Department of Mechanical Engineering, Baylor University, Waco, TX 76706, USA

\* Correspondence: sc_james@baylor.edu; Tel.: +1-(254)-710-2534

**Abstract:** The Naval Weapons Industrial Reserve Plant (NWIRP) in McGregor, Texas began manufacturing explosives in 1980 and several hazardous chemicals were discovered in lakes and streams surrounding the site in 1998. Contaminants traveled to local lakes and streams much faster than initially predicted. This research estimated contaminant travel times and identified locations where monitoring wells should be installed to yield the greatest reductions in uncertainties in travel-time predictions. To this end, groundwater and particle-tracking models for NWIRP site were built to predict hydraulic heads and contaminant travel times. Next, parameter (hydraulic conductivities) uncertainties, parameter identifiabilities, observation (hydraulic heads) worth, and predictive (contaminant travel times) uncertainties were quantified. Parameter uncertainties were reduced by up to 92%; a total of 19 of 158 parameters were at least moderately identifiable; travel-time uncertainties were reduced up to 92%. Additionally, travel-time predictions and post-calibration parameter distributions were generated using the null-space Monte Carlo (NSMC) technique. NSMC predicted that conservative tracers exited the flow system within a year, which matches with field data. Finally, an observations-worth analysis found that additional 11 more measurements would reduce travel-time uncertainties by factors from 1.04 to 4.3 over existing data if monitoring wells were installed at the suggested locations.

**Keywords:** parameter estimation; uncertainty analysis; forecasting; numerical modeling; Monte Carlo simulation; observation worth

## 1. Introduction

The Naval Weapons Industrial Reserve Plant (NWIRP) occupies about $40\,\text{km}^2$ in southwest McGregor, Texas on a topographic divide underlain by a shallow groundwater system within fractured limestone bedrock (Figure 1). The NWIRP began manufacturing explosives in 1980 [1,2] and stored several hazardous chemical wastes including ammonium perchlorate at five administrative locations: G, H, L, M, and S (see Figure 1) those were found in lakes and streams surrounding the plant in 1998 [3,4]. Such short travel times were inconsistent with estimates from groundwater velocities that suggested the contamination should not have entered streams and migrated offsite for decades [5]. Authorities failed on two fronts: (1) they significantly underestimated travel times for contaminants to reach nearby streams and (2) they failed to appropriately monitor the system after discovery of wastes exiting the site.

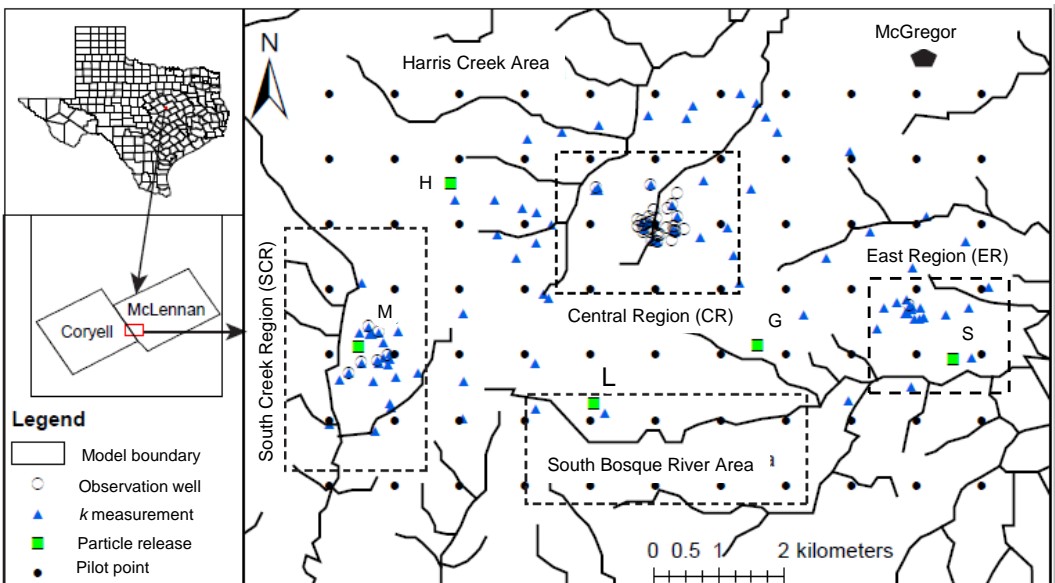

**Figure 1.** Study location showing streams, 43 observation wells (open circles), 99 hydraulic-conductivity measurement locations (blue triangles), 77 pilot points (filled circles), and particle-release locations (green squares) from NWIRP administrative designations G, H, L, M, and S.

To address these issues, a heterogeneous MODFLOW-NWT [6] model was developed to simulate flow and transport at the NWIRP site. The model included two fractured layers (the upper and lower layers), topography, streams, and heterogeneous hydraulic conductivities. In addition, a MODPATH [7] particle-tracking model estimated conservative tracer transport paths and transport times. Predictions were conditioned through calibration against 43 measured hydraulic heads. Uncertainties in these predictions were quantified and the most important parameters and observations identified.

Moreover, the Null-space Monte Carlo (NSMC) [8–10] nonlinear uncertainty analysis technique was used to generate probability distributions of calibrated parameters and commensurate particle travel-time predictions. Briefly, the NSMC technique uses subspace approaches like singular value decomposition [11] to identify only those model parameters informed by the observation dataset [12]. This facilitates inversion of over-parametrized models by only calibrating those variables about which the dataset has information while relegating the rest to their user-preferred initial guesses through Tikhonov regularization [13].

Based on linear and nonlinear uncertainty analyses using MODFLOW and MODPATH models of the NWIRP site, we quantified the times for particles to reach nearby streams, which ultimately feed into nearby rivers and lakes. We also suggested locations for additional monitoring wells and a future data collection plan to better characterize and monitor the site to support improved waste-containment operations. These additional data were selected to ensure the greatest reductions in model-prediction uncertainty.

*Background*

Ensafe Inc. [3] estimated groundwater volumetric flux in Georgetown Limestone at 2 m/year using average gradients, hydraulic conductivity from slug-test data, and a total porosity while assuming homogeneous and steady groundwater velocities throughout the area. However, those estimates did not consider the increased fluxes and hydraulic heads affecting groundwater flow velocities during storm periods. The Georgetown Limestone, similar to other fractured carbonates like the Austin Chalk, exists as an upper, highly fractured and unsaturated zone overlying a low-permeability, moderately fractured zone [14–18]. Recharge from precipitation significantly increases lateral flow through the Georgetown Limestone when the water table rises into the upper, highly fractured zone. The water table is sensitive to recharge (storms) as shown in Figure 2. A rising

water table can mobilize dissolved perchlorate such that it enters the upper zone where it is more easily transported off site. Even though some dissolved perchlorate is transported off site, the source persists as residual perchlorate in the lower fractured zone awaiting remobilization during the next storm (see Figure 3).

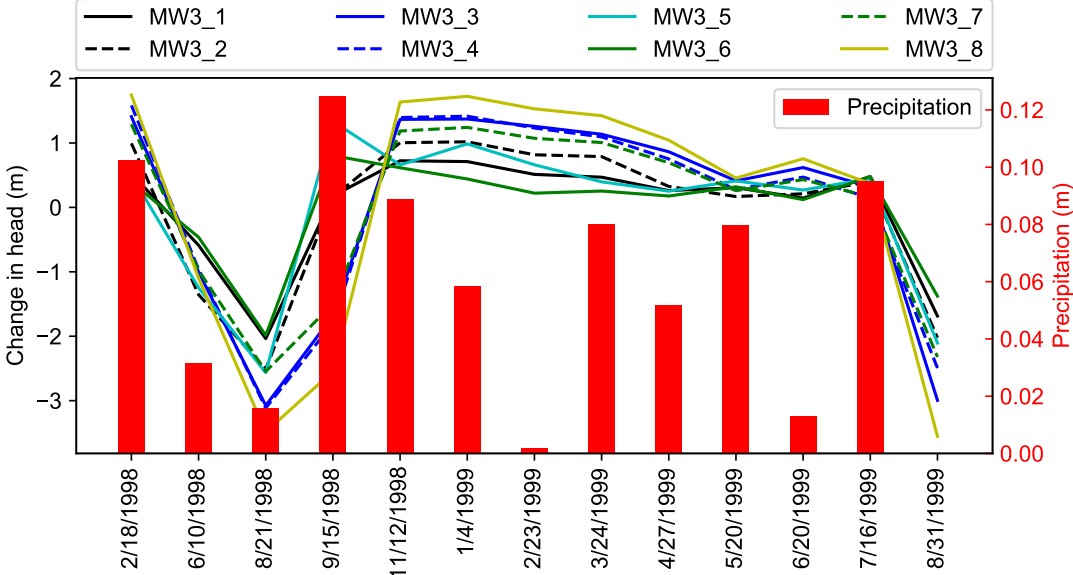

**Figure 2.** Change in head in response to precipitation over time. Here, hydraulic heads are plotted after removing their linear trends. Precipitation data: Waco Regional Airport [19].

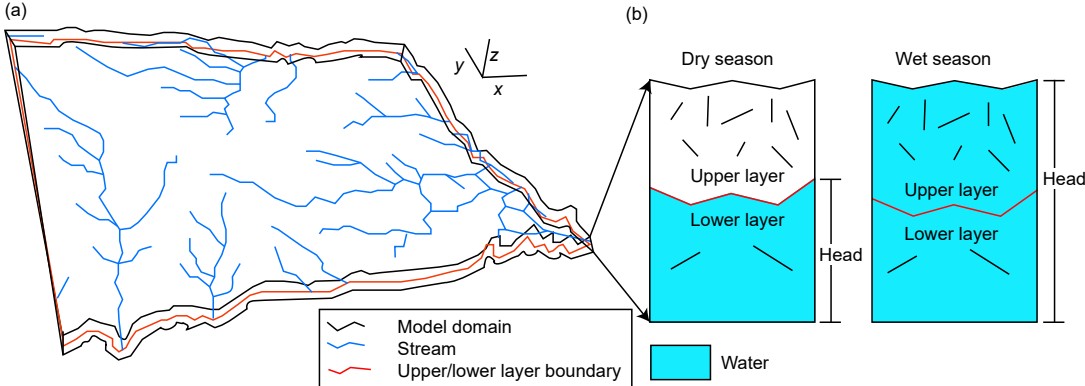

**Figure 3.** (**a**) Schematic of the model domain with undulating layers and streams; (**b**) Cross sections during dry seasons, contaminant travel slower in the less-fractured, lower layer and during the wet season, they travel faster in the more fractured, upper layer.

## 2. Modeling Approach

The level of parametrization of an environmental model should be commensurate with the quality and quantity of data used in its calibration to ensure confidence in the range of predictive possibilities [12,20]. Calibration is constrained by the information content of the calibration data set (plus expert judgment) and linear predictive uncertainty can be assessed even before a calibration exercise. Parameter uncertainty along with observation worth can be quantified [21,22]. Post-calibration, the NSMC method facilitates a nonlinear assessment of parameter and prediction uncertainties [22], but even with the use of super parameters (linear combinations of estimable parameters) to reduce the number of model calls, the approach can still be computationally intensive. The NSMC approach determines those parameters that are most informed by the calibration dataset

and focuses on calibrating those parameters. For parameters that are not strongly informed by the calibration dataset, regularization techniques maintain those parameters close to their user-specified values.

Predictive uncertainty analyses can be undertaken with a calibrated model using methods based on the propagation of variance [22], which acknowledges that historic observation can be replicated with many non-unique parameter combinations. Predictive uncertainty reduction is calculated based on the pre- and post-calibration parameter uncertainties, where pre-calibration parameter variances (uncertainties) are specified according to measurements and expert judgment while post-calibration uncertainties are revealed through the calibration process and commensurate post-calibration analyses. Reductions in uncertainties in model predictions result from the information content in the calibration dataset through the decrease in pre-calibration parameter uncertainties upon calibration. Solution-space uncertainty, usually the smaller of the two, is due to uncertainty in the calibration data (i.e., measurement error). Null-space uncertainty is due to shortcomings in the data or model that preclude precise identification of the parameter (i.e., many parameter combinations can calibrate the model about equally well). The mathematical process of distinguishing solution- from null-space uncertainty is achieved through singular value decomposition (SVD), which is conducted with straightforward mathematical vector and matrix manipulations [21,22].

Measurement errors (observation noise) can never be eliminated and these impact predictive uncertainties. The calibration process minimizes the weighted-sum-of-squares differences, the objective function, between site observations and their corresponding model predictions. Both quantitative (observation noise, measurement accuracy, number of measurements comprising an observation, etc.) and qualitative (expert judgment) metrics should be used to specify weights in the objective function.

Identifiability is a metric indicating the calibration data's ability to constrain a model parameter [23]. Quantitatively, it is the direction cosine between a parameter and its projection onto solution-space uncertainty. Identifiability can be used in both model design and implementation to assess whether a model needs more calibration data to reduce parameter uncertainty while also quantifying the uncertainties in predictions that depend specifically upon that parameter.

Observation worth is quantified based on the reduction in uncertainty in a parameter or prediction that is accrued through the acquisition of that data point [24]. Reduction in these uncertainties below their pre-calibration level is a measure of the worth of an observation (or observation group) with respect to that parameter or prediction.

The NSMC technique generates multiple, unique parameter fields that satisfy both the model-to-measurement misfit (i.e., a sufficiently low objective function) and parameter-reality constraints (i.e., parameters cannot be assigned unrealistic values) and it quantifies post-calibration parameter and prediction uncertainties [25]. It generates a suite of equally likely and realistic parameter fields that are used to make predictions. Generating parameter fields involves three steps: (1) generating random parameter fields according to pre-calibration uncertainty, (2) perturbing pre-calibrated parameters by removing the solution-space uncertainty, and (3) a brief model calibration with three optimization iterations using fewer parameters or (super parameters), which are linear combinations of those parameters that have their pre-calibration uncertainty reduced by a significant amount (user defined, but for example >10%). Uncertainty in a prediction can thereby be assessed through construction of an empirical probability density function (PDF) assembled by running the model using each NSMC parameter field realization to generate PDFs of predictions.

The NSMC approach is faster than the related Markov Chain Monte Carlo (MCMC) approach, which is a Bayesian technique that samples parameters/predictions to obtain posterior PDFs [22]. The MCMC approach requires two steps: Monte Carlo random sampling for all model parameters, not just those constrained by data, and Markov Chain sequencing of events that are probabilistically related. Essentially, MCMC methods approximate parameter posterior distributions by random sampling in a probabilistic space and then making the corresponding model runs. Similar to NSMC, a Bayesian analysis provides post-calibration parameter estimates with quantified probabilities.

For models with long run times and many parameters, MCMC becomes computationally expensive while Doherty [22] demonstrated that the additional model runs required by MCMC yield insignificant accuracy gains over NSMC.

*2.1. Conceptual Model*

The conceptual model was built using four digital elevation maps (DEMs) from the Texas Natural Resources Information System website [26] to create the model topography. GIS [27] and SURFER [28] were used to mosaic and grid the DEM data [29]. The conceptual model also included local rivers, streams, creeks, and spring as shown in Figure 3a. The model comprised two 4-m-thick layers representing the upper, weathered layer and the lower, less-permeable limestone. The NWIRP model domain and terrain-following layers were adjusted according to the topographic elevation (see Figure 3a). The top of the upper model layer was assumed 2 m below the land surface. The bottom of the model was established uniformly 10 m below surface [29]. Average precipitation from 1960 to 2015 was $2.7 \times 10^{-8}$ m/s. A base-flow study conducted in a nearby similar geologic setting indicated that 7% of total precipitation infiltrated to the aquifer [29,30], so recharge was assigned $1.8 \times 10^{-9}$ m/s. Due to lack of transient recharge and head data, a steady-state numerical model (see Section 2.2) was developed assuming that the upper layer was fully saturated during the storm periods; this upper layer was, on average, five orders of magnitude more conductive than the bottom layer.

*2.2. Numerical Model Development*

The upper layer in the model domain is densely fractured (Figure 4); however, the fracture system has not been characterized for this site (e.g., no data on fracture length, spacing, aperture, orientation, etc.). A dense fracture system significantly enhances hydraulic conductivity of a rock formation. Sufficient data were available to characterize the system as an equivalent porous medium with enhanced hydraulic conductivity for the upper layer compared to the lower layer. Therefore, this effort started by developing groundwater and particle tracking models of the NWIRP site (Figure 1), calibrating to available data, assessing observation worth, as well as quantifying uncertainties in parameters and particle travel-time predictions. Next, results from the NSMC approach were used to generate PDFs of pilot-point hydraulic conductivities and travel times. A grid-refinement study was performed to adopt an optimal grid resolution for this study. Cell dimensions along x and y axes for the grid-refinement study were $20 \times 20$ m$^2$, $25 \times 25$ m$^2$, $50 \times 50$ m$^2$, and $100 \times 100$ m$^2$, each with two layers. Figure 5 shows that finer grids increased run times with negligible accuracy changes for piecewise homogeneous hydraulic conductivities. Therefore, we considered the model with two layers (representing the highly conductive upper layer overlying the less-conductive lower layer), 126 columns, and 97 rows with $100 \times 100$ m$^2$ cells.

Several boundary conditions (BCs) were applied in the model domain. Recharge through precipitation was specified at the top of the model while the lateral sides along with the bottom of the model were specified as no-flow boundaries. No-flow boundaries can affect results near the edges of the model, so the NWIRP regions of interest were always at least 1.5 km (15 cells) from the model edges. In addition to specified-flux and no-flow BCs, head-dependent-flux (HDF) BCs were assigned to cells containing a stream (drain cells in MODFLOW). In the HDF BC, a reference head and a conductance value were assigned at stream boundaries where water may only leave the flow system through the drain cell and it may not re-enter the groundwater system through cells further downstream. Also, if the hydraulic head in the stream cell fell below a certain threshold (e.g., bottom of the upper layer), the flux from the drain to the model cells was set to zero. The model was executed with MODFLOW-NWT [6], which admits the drying and re-wetting nonlinearities of an unconfined aquifer [6,31]. Single particles were released at the midpoint of each layer from the five administrative areas and tracked until they exited the model domain at streams. Streams were modeled as potential discharge cells.

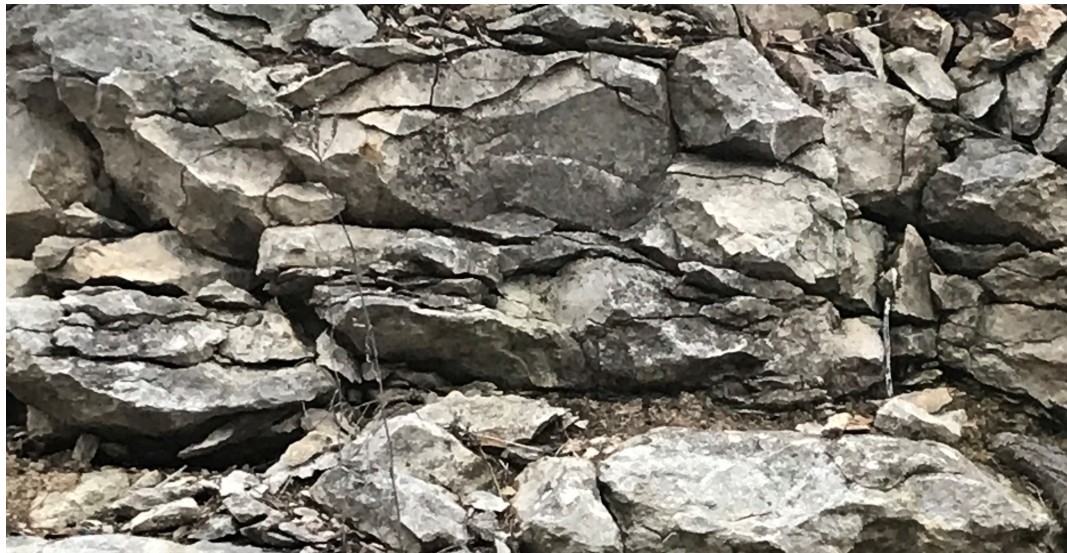

**Figure 4.** Picture of the fractured formation in the study area (scale is 0.5 m$^2$).

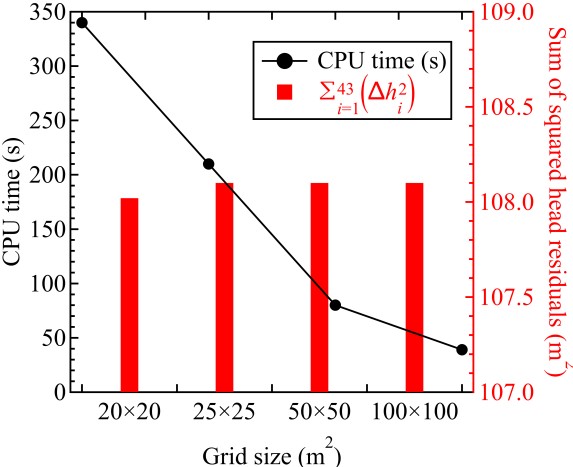

**Figure 5.** The black curve shows the time taken to run a single simulation across different grid resolutions. The bars indicate the squared difference between observed and simulated hydraulic heads and they are all about equal for piecewise homogeneous hydraulic conductivities.

*2.3. Parameters*

Groundwater flow is fundamentally governed by the distribution of hydraulic conductivities. In conventional calibration methods, property uniformity or pilot-point distributions are used as the basis for spatial parameter distribution [32]. In the absence of data, piecewise-homogeneous zones are often specified. If geologic zones are not piecewise-uniform, pilot points are distributed throughout such zones. Pilot-point property values were estimated during this calibration exercise and the hydraulic conductivities at model cells were assigned according to a kriging algorithm [32]. Pilot points facilitate a smooth but plausible distribution of hydraulic properties over a geologic unit, which cannot be achieved using piecewise-uniform methods. The upper model layer had only a single hydraulic conductivity measurement, but one parameter over such a large region would lend false confidence in the solution because it would yield unrealistic homogeneity [33]. Instead, a total of 77 pilot points (Figure 1) were used in each layer such that hydraulic conductivity fields were developed with heterogeneity commensurate with the information available in the observation data set. The initial value of the horizontal hydraulic conductivity in the upper layer, $k_\mathrm{u}$, was $3.048 \times 10^{-3}$ m/s for all 77 pilot points with a 1.5% porosity [29].

Ensafe Inc. [3] conducted 99 slug tests estimating horizontal hydraulic conductivities in the lower layer, $k_l$, ranging from $10^{-8}$ to $10^{-4}$ m/s with mean $10^{-7}$ m/s. These hydraulic-conductivity estimates were used in an exponential variogram with specified range and sill (variance) [34]. The range and variance of log of hydraulic-conductivity measurements were 700 m and 1.52, respectively, and using the 99 measured hydraulic conductivities, they were kriged (interpolated) onto each model cell. The vertical anisotropies and porosity of the lower layer were specified as one tenth of horizontal hydraulic conductivities and 0.5% [29], respectively. Note, although it could facilitate a higher level of heterogeneity, no nugget effect was considered in this study because the empirical semi-variogram did not require it. Nevertheless, our broad hydraulic conductivity constraints during model calibration effectively interrogated a broad uncertainty range and admitted a high degree of heterogeneity.

The calculated range and sill were used to generate kriging factors for pilot points in the lower layer. Later, these factors were used to interpolate $k$ onto the model grid using kriging. Because of exposure to weathering and erosional process, the upper layer is more heterogeneous even though it comprises similar rock types, so a larger variance was appropriate. Thus, a variance of 3.04 (almost twice that of the lower layer) was assigned to the upper layer with the same 700-m range as the lower layer. Variances of horizontal anisotropies for the upper and lower layers were set to 0.61. Parameter uncertainties and observation worth were calculated based on propagation of variance. Initially, a pre-calibration covariance matrix was calculated for the pilot points. The diagonal elements of the covariance matrix were the $k$ variances while off-diagonal elements were non-zero covariances based on the 99 estimated hydraulic conductivities and their geospatial characteristics. This covariance matrix was used to generate pre-calibration pilot point realizations.

A total of 158 log-transformed parameters were adjusted during calibration. Parameters were subdivided into four groups: (1) 77 pilot-point-based horizontal hydraulic conductivities for the upper $(k_{u_1}-k_{u_{77}})$, (2) and lower $(k_{l_1}-k_{l_{77}})$ layers, (3) horizontal anisotropies for the upper, $h_u$, and lower, $h_l$, layers, and (4) vertical anisotropies for the upper, $v_u$, and lower, $v_l$, layers. Each hydraulic conductivity was assigned a pre-calibration lognormal probability distribution with mean of $3.048 \times 10^{-3}$ and $10^{-7}$ m/s for the upper and lower layers, respectively.

### 2.4. Calibration Data and Predictions of Interest

Because of model complexity and under-determinacy, regularization was used during calibration [22] to reduce bias and to decrease the required number of model calls. The calibration was performed against 43 steady-state hydraulic heads (average hydraulic head if multiple measurements were available) at the monitoring wells indicated with open circles in Figure 1 (see, Appendix A for hydraulic-head data). Each observation was assumed equally important (equal weight). Predictions of interest were travel times for 10 particles released at the midpoints of the upper and lower layers at the five administrative hazardous storage sites labeled in Figure 1 [2]. Travel times for the particles were simulated and their uncertainties were quantified.

### 2.5. Calibration and NSMC

Steps in the calibration and uncertainty analyses included: (1) parametrization and calibration of the NWIRP groundwater model; (2) observation worth were calculated; (3) the effects of additional monitoring wells on travel-time predictions and observation worth were estimated; (4) uncertainties in travel-time predictions were then explored upon consideration of the new hypothetical observations (i.e., future borehole locations); and (5) post-calibration uncertainties of pilot points and particle travel times were assessed using the NSMC technique.

With 158 adjustable parameters (154 pilot points, two horizontal anisotropies, and two vertical anisotropies) and only 43 observations, this made this an ill-posed problem; not all parameters could be uniquely estimated [11,35]. Using SVD, it was determined that 25 unique linear combinations of parameters (super parameters) could be reasonably identified. In other words, all parameters (113 more than could possibly be identified by the 43 data points) were presented to the calibration and

PEST's subspace-regularization capabilities determine which parameters can be uniquely identified by the dataset and subsequently only focuses on those. Here, the dataset informed 25 parameters, which can be specified as linear combinations if there are strong correlations between some of them. For over-parametrized models, an NSMC analysis affords a more accurate, nonlinear assessment of predictive uncertainty. The NSMC approach was executed in three steps. First, a total of 1000 pilot-point realizations were generated (the mean of each pilot point stabilized by 1000 realizations) using a lognormal distribution with mean (from pump tests) and the pre-calibration covariance matrix. Second, each pre-calibrated parameter realization was perturbed by adding null-space uncertainty. This step calculated the orthogonal differences between calibrated parameters and random parameter realizations. Then, these differences were added to each realization of calibrated parameters to generate 1000 realizations of calibrated parameters with perturbations in the null space. For each of these null-space-projected parameter-field realizations, three iterations of a parameter calibration were undertaken perturbing pre-calibrated parameters by removing the solution-space uncertainty. Model results (hydraulic heads and particle travel times) using the newly generated parameter fields were necessarily similar to the results from the calibrated model; however, up to three calibration iterations (each yielding an updated Jacobian matrix and set of calibrated parameters) were undertaken to see if these steps could bring the model back into calibration. If a sufficiently low objective function could not be achieved after these three calibration iterations, then that parameter field was discarded (as not effectively calibrating the model or maintaining parameter reality).

## 3. Results and Discussion

### 3.1. Post-Calibration Linear Uncertainty Analyses

Figure 6 compares measured (*y* axis) and calibrated (*x* axis) hydraulic heads. There was a slight bias toward underprediction ($-0.1$ m) while the mean absolute error was 0.7 m and the root-mean-squared error 0.8 m, all of which indicate a well-calibrated model.

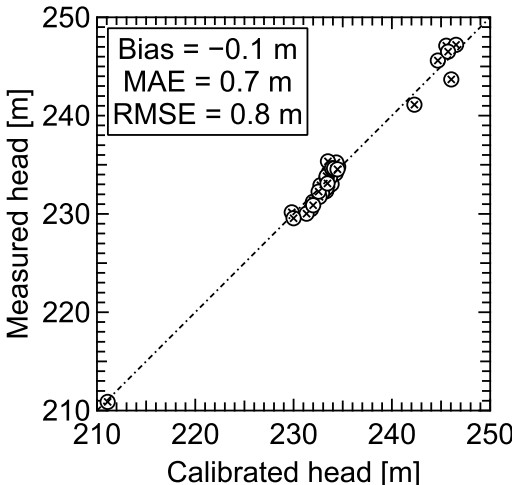

**Figure 6.** Cross-plot of measured and calibrated hydraulic heads.

The linear uncertainty analysis quantified the decreases in parameter and prediction uncertainties (i.e., variances of the pilot-point hydraulic conductivities and variances in particle travel times, respectively) subject to the information content in the calibration dataset. Post-calibration parameter uncertainties reflect the degree to which the observations decreased the pre-calibration parameter uncertainties as indicated in Figure 7. The size and color of the circles indicate the percent reduction in pre-calibration uncertainty upon application of the observation data set. Pilot-point uncertainty reductions ranged from 0.4 to 92% with greater reductions nearer to observation wells in the upper layer (Figure 7a). The Central (CR), Station Creek (SCR), and East Regions (ER) of the model, indicated on

Figure 1, contain 36, six, and one observation well(s), respectively. In the upper layer, observation wells in the CR significantly informed the six nearby pilot points by reducing their uncertainties from 15 to 92%. Similarly, the six wells in the SCR decreased uncertainties in four nearby pilot points from 25 to 90% while the single well in the ER notably decreased uncertainty in the nearest pilot point by 50%. Uncertainty reduction for pilot points in the lower layer ranged from 0.3 to 30%. Observation wells in the CR reduced uncertainties from 1 to 30%; the uncertainty reductions were relatively lower in this layer because of their smaller (by half) variance 1.52 ($1.78 \times 10^{-10} \, \mathrm{m^2/s^2}$). The observation wells in SCR informed three nearby pilot points and reduced their uncertainties by >10% (Figure 7b). All told, uncertainty reductions for 120 of the pilot points were <10%, typically for pilot points distant from observation wells. Overall, given that there were 43 unequally distributed monitoring wells and low initial uncertainties in the lower-layer pilot points, it was not surprising that only 36 pilot points had their uncertainties reduced by >10%. For the horizontal anisotropies, $h_u$ was significantly constrained by the calibration dataset, which reduced $h_u$ variance from 0.61 to 0.05 ($\approx$92% reduction). However, $h_l$ was not constrained by the calibration dataset. Like $h_l$, vertical anisotropies ($v_u$ and $v_l$) were not constrained by the calibration dataset.

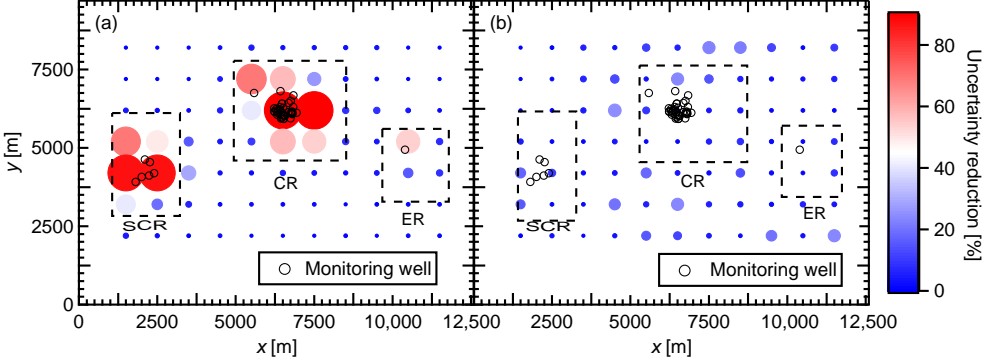

**Figure 7.** Reductions in pre-calibration uncertainties for the (**a**) upper- and (**b**) lower-layer pilot points upon application of the observation data set (43 head measurements).

Parameter identifiability apportions parameter uncertainties into solution and null spaces. An identifiability of zero means that the data set says nothing about that parameter while an identifiability of one means that uncertainty in that parameter is solely due to measurement error. Of the 158 parameters, seven had high (>0.5), 12 moderate (0.1–0.5), and 139 low (<0.1) identifiabilities (Figure 8).

Initial travel-time variances were calculated by running the 1000 uncalibrated hydraulic-conductivity-field realizations and these uncertainties were reduced by up to 92% (i.e., standard deviation in travel times reduced from 3.15 to 0.25 years) when the model was run with the calibrated hydraulic-conductivity-field realizations (discussed below in the Contaminant Travel-Time Prediction using NSMC subsection). Next, an observation-worth analysis was performed to assess the contribution of monitoring wells toward reducing travel-time uncertainties. Normalized contributions to uncertainty reduction from each monitoring well are indicated in Figure 9a. The contributions of observations toward decreasing travel-time uncertainties depended upon the degree to which parameters were constrained along the particle paths and the proximity of the observations to the particle paths. Monitoring wells in the SCR reduced uncertainties for particles released at M and L while one monitoring well in the CR reduced uncertainties for particles released at H and one monitoring well in the ER reduced uncertainties for particles from S. Particles released at G did not travel through a region with monitoring wells, so their uncertainties were not reduced. Overall, only four monitoring wells (large open circles in Figure 9a) significantly contributed to decreasing travel-time uncertainties indicating that the existing monitoring well network does not effectively constrain contaminant transport times.

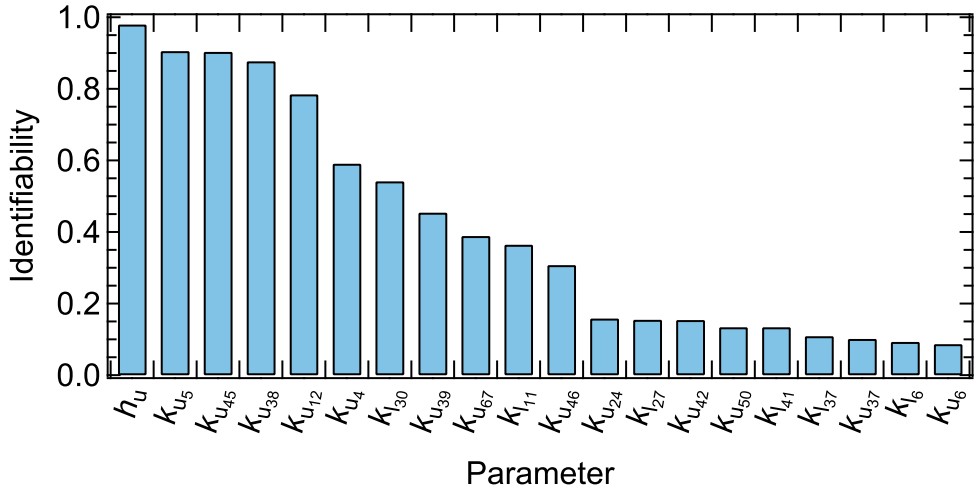

**Figure 8.** The 20 most identifiable parameters.

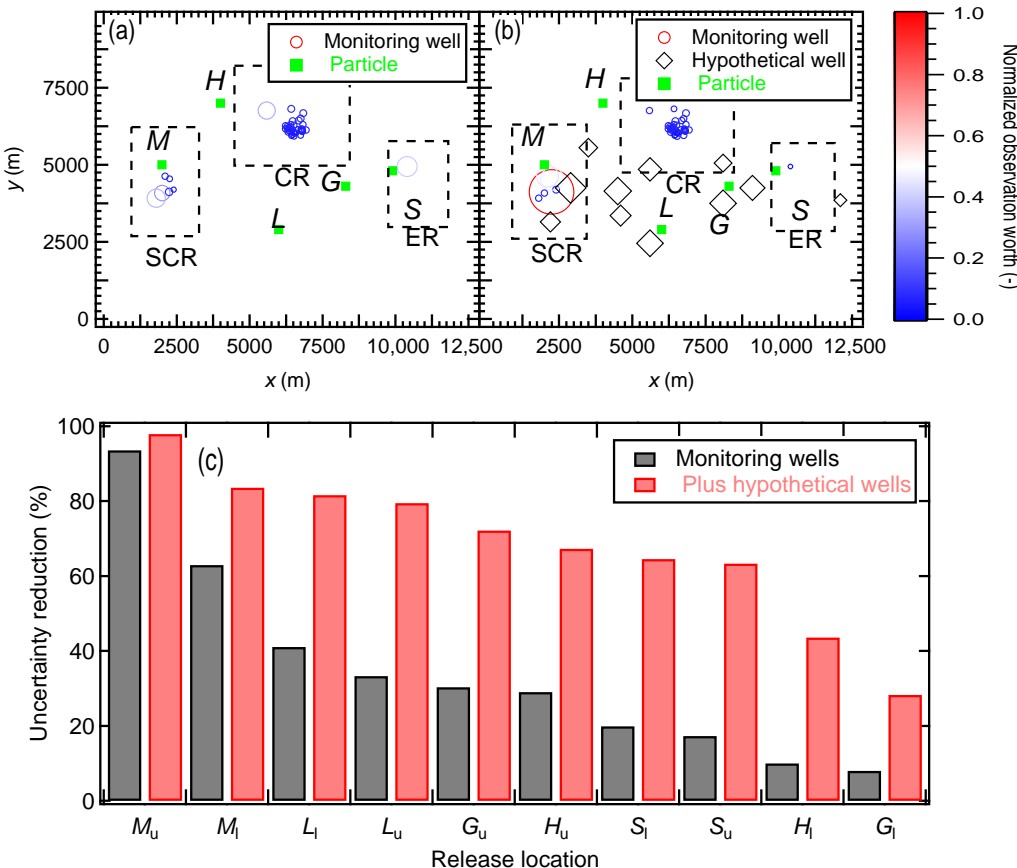

**Figure 9.** Normalized observation worth of (**a**) monitoring wells (**b**) plus 11 hypothetical wells, symbol color and size indicate the observation worth; and (**c**) uncertainty reductions in travel times due to the existing wells (gray) and as augmented by 11 hypothetical wells (red). Subscripts u and l represents upper and lower layers, respectively.

### 3.2. Contaminant Travel-Time Prediction Using NSMC

Of the 1000 parameter-field realizations generated, 882 had objective functions less than 1.5 times the calibrated (minimum) objective function after three optimization iterations of an NSMC re-calibration. These calibration-constrained NSMC realizations and the corresponding random

882 realizations were used to compute parameter variances to assess the decreases in uncertainty due to the information content in the calibration data set. Figure 10 compares the uncertainty reductions according to the linear (blue curves) and nonlinear (NSMC, red curves) to *a priori* uncertainties (black lines). In the upper layer, all parameters showed at least some nonlinear uncertainty reduction. There was also a clear trend where 14 parameters in the upper layer were notably informed by the calibration dataset (variances of $\log_{10} k < 2$ (log-m$^2$/s$^2$)) regardless of the approach. For the bottom layer, the nonlinear uncertainty reductions tended to be greater than their linear counterparts with more parameters seeing significant variance decreases. The numerous model runs required by the NSMC approach yielded the additional variance reductions.

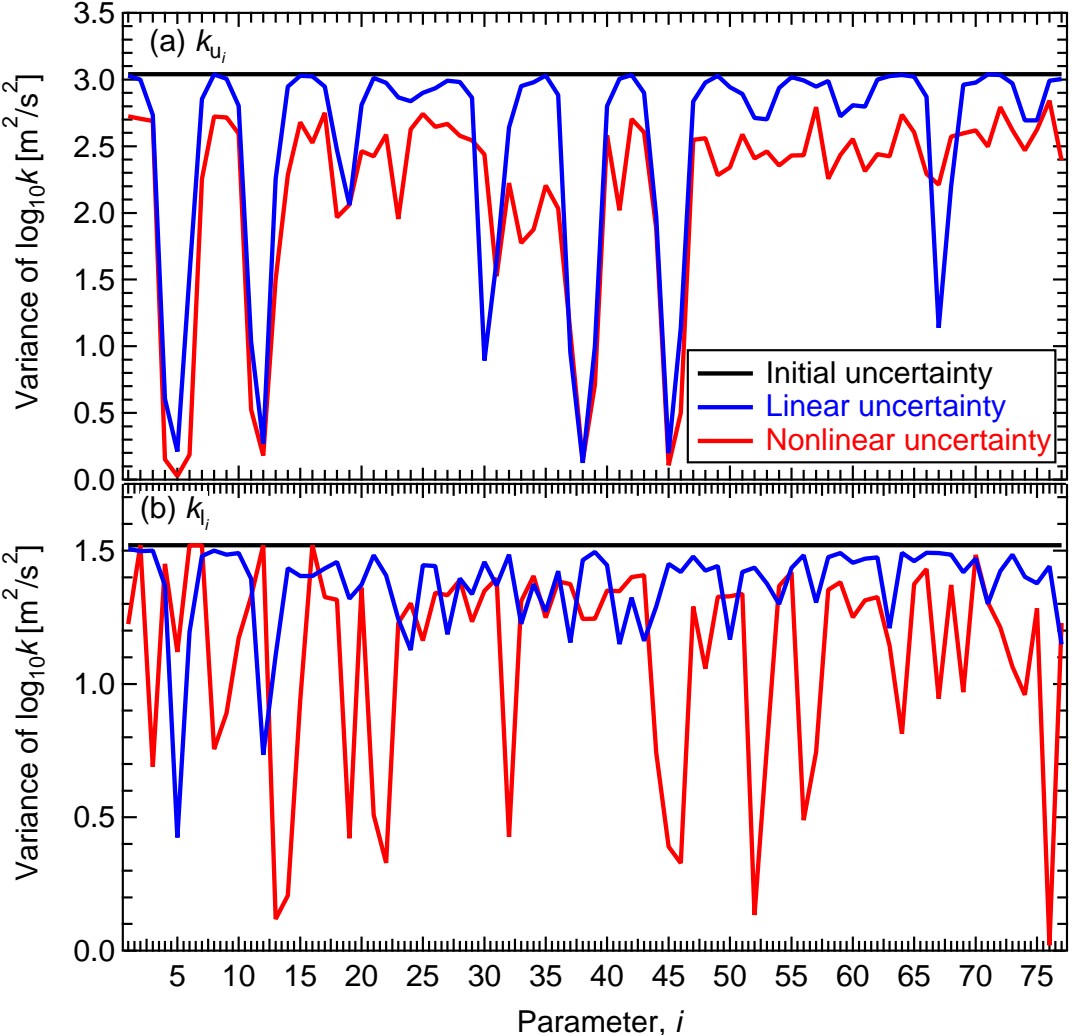

**Figure 10.** Variances of pilot-point $\log_{10}(k)$ for the (**a**) upper and (**b**) lower layers according to the linear (blue) and nonlinear (NSMC) uncertainty quantification techniques. The black lines indicate the *a priori* uncertainties.

Parameter realizations were also used to generate PDFs and Figure 11 provides six examples. For highly identifiable parameters (e.g., $k_{u_5}$ and $k_{u_{45}}$) in the top row of Figure 11, distributions were significantly narrower for NSMC than from pre-calibration reflecting the information obtained from the null-space projection and three additional optimization iterations. Distributions of the moderately identifiable parameters (e.g., $k_{u_{39}}$ and $k_{u_{67}}$) in the middle row of Figure 11 were also narrower than their pre-calibration equivalents. Finally, the three optimization iterations slightly reduced uncertainties for

minimally identifiable parameters (e.g., $k_{l_{37}}$ and $k_{l_{41}}$) in the bottom row of Figure 11, because those parameters were not informed by existing monitoring wells.

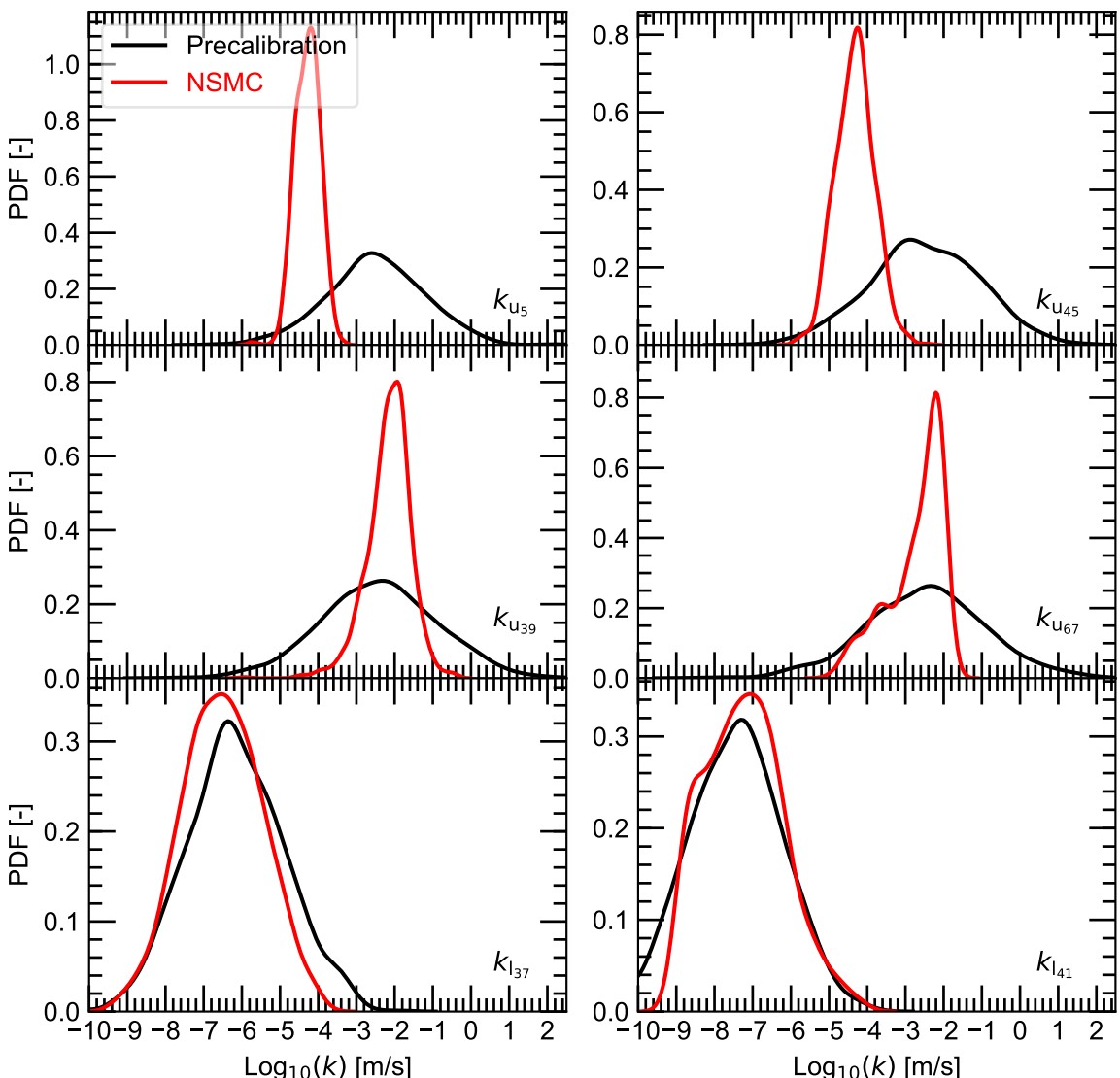

**Figure 11.** PDFs of highly identifiable parameters $k_{u_5}$ and $k_{u_{45}}$ (top row), moderately identifiable parameters $k_{u_{39}}$ and $k_{u_{67}}$ (middle row), and minimally identifiable parameters $k_{l_{37}}$ and $k_{l_{41}}$ (bottom row).

Travel times for particles to discharge at surrounding streams were calculated for each of the 882 pre-calibration and NSMC calibration-constrained pilot-point parameter fields and Figure 12 shows distributions of log-transformed travel times. Figure 13 shows 882 particle tracks released from each administrative designation with a representative hydraulic conductivity field. A particle's travel time depended on distance traveled and hydraulic conductivities along its path. Passing through even a single low-hydraulic-conductivity cell significantly decreased that particle's travel time. The combination of one or more low-hydraulic conductivities along a particle's path in conjunction with lognormally distributed hydraulic conductivities (long tails toward low values) yielded some realizations with exceptionally long travel times, hence it was more appropriate to compare median travel times (Table 1). Note that particles may only exit the model through stream cells; therefore, for both layers, particles discharged into streams (see Figure 13). Although not shown, exit points for the pre-calibration particle tracks were similar to those from the NSMC realizations except that they spanned a wider distance along the streams. The pre-calibration particle tracks are not presented because they are much less important than quantifying changes to travel times.

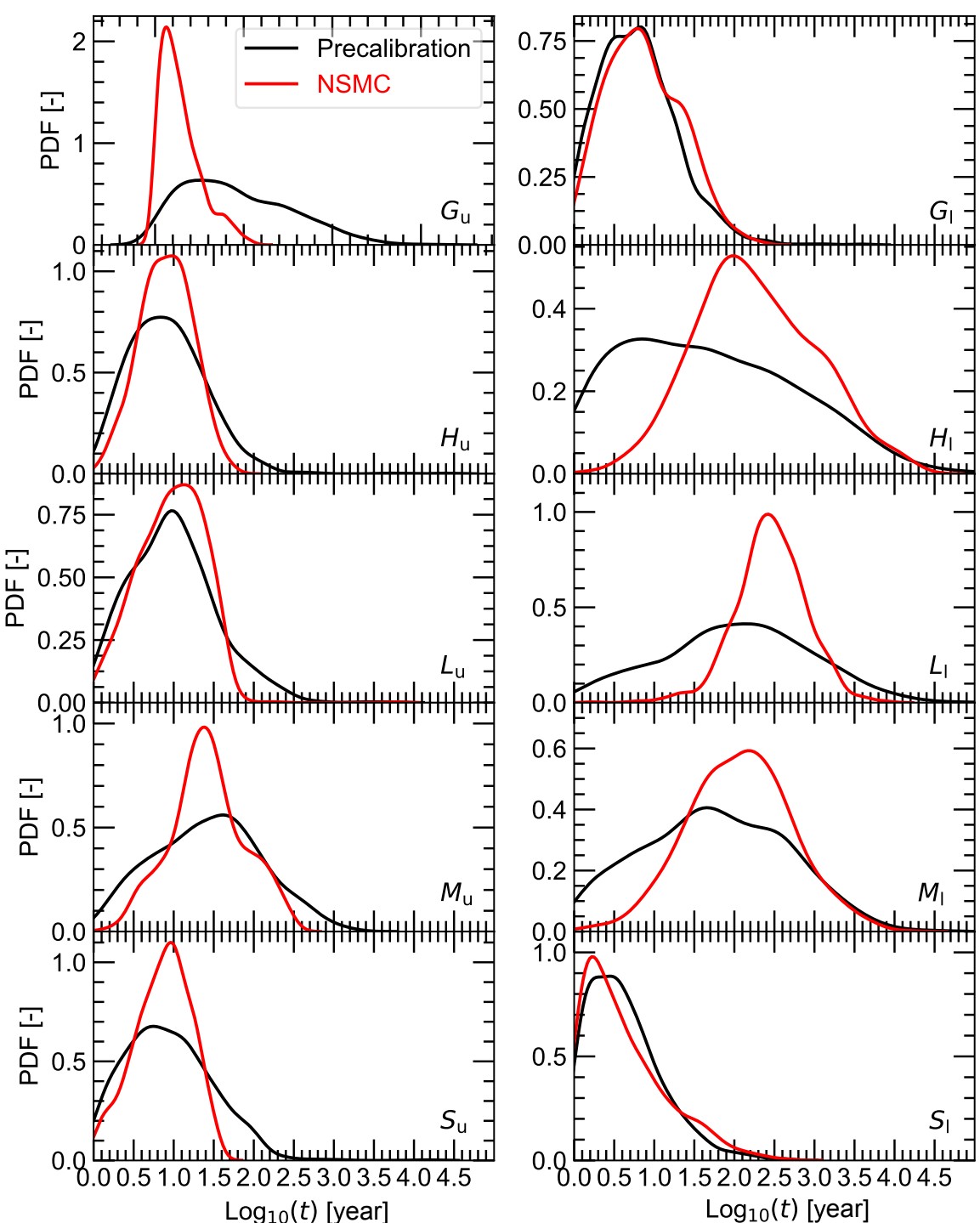

**Figure 12.** PDFs of particle travel times.

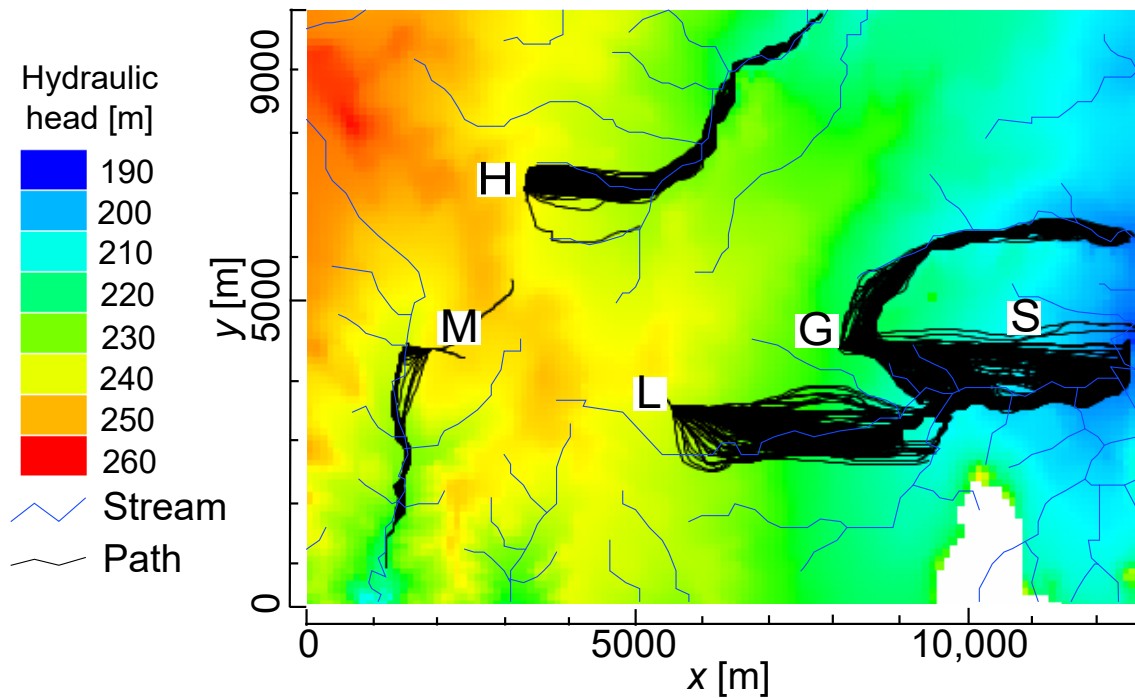

**Figure 13.** Particle paths for 882 realizations with background colored according to a representative hydraulic-head distribution (the white region in the southeast is an inactive portion in the model domain). Dark red semicircles represent suggested barriers to prevent contaminant spreading.

**Table 1.** Median travel times for the 882 pre-calibration and NSMC parameter fields used for particle tracking. Upward arrows indicate an increase and vice versa for the downward arrows.

| Particle | Pre-Calibration [Year] | NSMC [Year] |
|---|---|---|
| $G_u$ | 0.29 | ↑ 0.65 |
| $G_l$ | 5.51 | ↑ 6.65 |
| $H_u$ | 0.14 | ↓ 0.12 |
| $H_l$ | 39.01 | ↑ 166.34 |
| $L_u$ | 0.11 | = 0.11 |
| $L_l$ | 109.92 | ↑ 303.23 |
| $M_u$ | 0.03 | ↑ 0.04 |
| $M_l$ | 54.67 | ↑ 122.98 |
| $S_u$ | 0.13 | ↓ 0.12 |
| $S_l$ | 0.96 | ↑ 2.36 |

Particles released at the same location for pre-calibration and NSMC parameter realizations had similar path lengths in both layers, but median travel times in the upper layer were significantly shorter than those through the lower layer. Consistent with the conceptual model, hydraulic conductivity ranges were two to six orders of magnitude higher in the upper layer than the lower layer resulting in the travel-time disparities. However, all simulations indicated that particles released in the upper layer reached surrounding streams within a year (consistent with site observations) while particles released in the lower layer took one to five orders of magnitude longer.

Although travel times through the lower layer were longer when using pre-calibration parameter realizations than their NSMC counterparts, there was no consistent trend for the upper layer. Nevertheless, corresponding travel-time uncertainties (variances of log-transformed travel times) always decreased from pre-calibration to NSMC (visually evident in Figure 12), which is consistent

with the decreases in parameter variances (see Figure 10). Although the NSMC travel times through the lower layer indicated that there may be more time to remediate the contamination than might have been initially expected, the short travel times through the upper layer offered no such consolation.

*3.3. Suggestion for Future Data Collection Points*

Additional targeted well installations would greatly reduce travel-time uncertainties. To this end, an observation-worth analysis was performed by adding hypothetical monitoring wells requiring only dummy hydraulic-head values. The analysis was conducted by placing wells at every third model cell throughout the model domain and selecting optimal locations (i.e., those that yielded the greatest reductions in uncertainty). The resulting 11 wells (Figure 9b) would be drilled down gradient of particles released at administrative area M and H and in the vicinity of hazardous-materials storage sites L, G, and S where no wells exist. The normalized contributions of monitoring plus hypothetical wells in reducing travel-time uncertainties are indicated in Figure 9b. These 11 hypothetical wells were located where they yielded the greatest reductions in travel-time uncertainties. Interestingly, upon adding the 11 hypothetical wells, the contribution of an existing well toward reducing travel-time uncertainty greatly increased (largest red circle) because it gained important gradient information (i.e., one of the hypothetical wells provided the second water-level necessary to calculate the gradient between the new and old well, thereby yielded information constraining the hydraulic conductivity).

Post-calibration sensitivities of travel-time predictions to observations were calculated and the degree to which observations reduced uncertainties in travel times are indicated in Figure 9c. $M_u$ and $M_l$ had six nearby monitoring wells and these particles traveled through a well-constrained region of the model and happened to have the shortest travel distances; hence uncertainties were notably reduced by the existing wells. The addition of the 11 hypothetical wells further constrained parameters (and corresponding predictions) such that travel-time uncertainties for particles released at H, G, L, and S decreased by factors from 1.04 to 4.3 (red bars) compared to the existing well network.

## 4. Conclusions

Based on groundwater and particle-tracking simulations of the NWIRP site, calibration and model interrogation quantified parameter and predictive uncertainties, parameter identifiabilities, and observation worth for both existing and hypothetical monitoring wells. Using a linear analysis, pre-calibration parameter uncertainties were reduced up to 92% and 36 of 158 parameters exceeded a 10% reduction when constrained by the calibration data set. An identifiability study revealed that seven parameters were highly identifiable (>0.5) while 12 parameters had identifiabilities between 0.1 and 0.5. Travel-time uncertainties were reduced up to 92%. Using a nonlinear analysis, pre-calibration travel-time uncertainties were reduced by >50% for two particles released at site M and between 5 and 40% for the remaining sites. Vertical anisotropy did not play a significant role because there was minimal vertical flow in the model and no well data suggesting a vertical gradient.

Also, this study generated pre- and post-calibration parameter distributions along with corresponding travel-time PDFs. The decreases in width of these distributions (variances) reflected the information content in the calibration data set. This study also predicted travel times for conservative particles to reach nearby streams and, for the upper layer, they exited the flow system within a year. An observation-worth analysis showed that the existing monitoring well network does not strongly constrain travel times. Suggested data collection, especially at the locations shown in Figure 9b, could reduce travel-time for all particles by factors from 1.04 to 4.3. Ultimately, it is up to the regulator to determine how many wells must be added to reduce modeling uncertainties to a point where they are comfortable making remediation decisions based on simulated site behavior.

Any seriously contaminated site like NWIRP should undergo a rapid and detailed modeling study before further data collection and, of course, before making remediation decisions. For example, the authority collected clustered water-level measurements, which could have been optimized if a comprehensive study was conducted before drilling wells. In addition, no base-flow data

were collected even though a single base-flow measurement significantly improves uncertainty quantification [36]. This study also demonstrated that if a contaminant reached the upper layer (for example during storm events that raise the water table), it will travel much faster to surrounding streams. The predictive-uncertainty and observation-worth analyses determined the most important parameters and observations contributing to the greatest decreases in predicted travel-time uncertainties. Ultimately, the 43 poorly distributed water-level measurements over such a large model domain and the absence of base-flow data were notable shortcomings. This analysis can support decision making by identifying where additional wells should be located to achieve the greatest reductions in predictive uncertainty.

Looking to the future, transient modeling would be appropriate for this system, but it cannot currently be undertaken because of a lack of water-level time-series data. Although beyond the scope of this study, it would be reasonable to use these calibrated parameters in a transient model to assess other aspects of contaminant transport subject to storm or flood events. While a discrete fracture network model could be an alternative approach, the fractured system was not sufficiently characterized to support development of such a model. Also, chemical wastes were treated as conservative particles. Although a more complex transport model would also be of value, the necessary data (e.g., a comprehensive study on the chemical wastes including measurements of dispersion, matrix diffusion, sorption, decay, and other local or regional mixing phenomena for individual chemical constituents) are not available.

**Author Contributions:** Simulation run, B.A.; manuscript draft, B.A.; conceptualization, J.Y.; software, S.C.J.; supervision, S.C.J. All authors have read and agreed to the published version of the manuscript.

**Funding:** This research received no external funding.

**Conflicts of Interest:** The authors declare no conflict of interest.

## Appendix A

**Table A1.** Locations and average heads of observations.

| Observation | $x$ (m) | $y$ (m) | Head (m) | Observation | $x$ (m) | $y$ (m) | Head (m) |
|---|---|---|---|---|---|---|---|
| MW3-1 | 6422 | 6072 | 233.65 | GAF-2 | 6832 | 6306 | 232.26 |
| MW3-2 | 6407 | 6230 | 232.66 | GAF-3 | 6940 | 6124 | 233.09 |
| MW3-3 | 6357 | 6206 | 232.30 | GAF-4 | 6749 | 5949 | 235.23 |
| MW3-4 | 6346 | 6173 | 232.44 | GAF-5 | 6577 | 6027 | 234.64 |
| MW3-5 | 6456 | 6142 | 233.47 | GAF-6 | 6512 | 6045 | 234.52 |
| MW3-6 | 6458 | 5975 | 234.12 | GAF-7 | 6426 | 6809 | 230.17 |
| MW3-7 | 6326 | 6120 | 232.83 | GAF-8 | 6755 | 6498 | 230.02 |
| MW3-8 | 6302 | 6177 | 232.31 | GAF-9 | 6835 | 6673 | 229.55 |
| MW3-9 | 6291 | 6238 | 232.29 | AF0MW01 | 6668 | 6135 | 235.35 |
| MW3-10 | 6526 | 5924 | 234.80 | AF0MW02 | 6539 | 6054 | 234.65 |
| MW3-11 | 6315 | 6064 | 233.03 | AF0MW03 | 6543 | 6012 | 234.63 |
| MW3-12 | 6513 | 6283 | 232.93 | AF0MW04 | 6439 | 5940 | 234.54 |
| MW3-16 | 6409 | 6307 | 231.74 | MW1-3 | 10396 | 4946 | 210.89 |
| MW8-1 | 6768 | 6108 | 234.22 | MW2-1 | 2001 | 4079 | 245.60 |
| MW8-2 | 6801 | 6102 | 234.03 | MW2-4 | 1804 | 3916 | 241.10 |
| MW8-3 | 6808 | 6097 | 234.03 | MW4-3 | 5587 | 6757 | 230.89 |
| MW8-4 | 6807 | 6079 | 233.70 | MW5-1 | 2397 | 4190 | 243.69 |
| MW8-5 | 6802 | 6151 | 233.70 | MW5-2 | 2239 | 4119 | 247.10 |
| MW8-6 | 6771 | 6151 | 233.84 | MW7-1 | 2264 | 4547 | 247.19 |
| GAF-1 | 6685 | 6433 | 230.49 | MW7-3 | 2107 | 4627 | 246.50 |
| MW3-13 | 6231 | 6164 | 232.65 | MW3-14 | 6228 | 6263 | 232.10 |
| MW3-15 | 6474 | 6417 | 231.23 | | | | |

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
