# Peer review of "Post-Calibration Uncertainty Analysis for Travel Times at a Naval Weapons Industrial Reserve Plant"

_water, doi:10.3390/w12123428_

Round 1

Reviewer 1 Report

This paper study the contaminant travel times and capture zones of a site called NWIRP situated in Texas. The objective is to identify locations where monitoring wells should be installed to reduce possible contamination.
The method used is a heterogeneous MODFLOW-NWT model to simulate flow and transport together with MODPATH to determine the capture zones, and contains two fractured layers and heterogeneous hydraulic conductivities. They also used a NSMC nonlinear uncertainty analysis technique to generate probability distributions of the calibrated parameters and particle travel-time predictions.

In the paper, some issues have to be better explained/corrected:

line 129: It is a good approximation to relate rock densely fractured to a porous medium?

line 135: I do not understand how many cells contains the model. Is it 2 (layers) x 126 columns x 97 rows? then, each cell has an area of 100 x 100 m^2? this is a huge area for a single cell in the grid and I am not sure that results might be reliable.

line 127: a constant value of the recharge is not a good approximation for such a complex site.

line 344: "This study also demonstrated that if a contaminant reached the upper layer (for example during 342 storm events that raise the water table), it will travel much faster to surrounding streams." I do not see how this might be possible if the recharge was set to be a constant value and there is not a water-level time series data (line 349).

Author Response

Comments and Suggestions for Authors

This paper studies the contaminant travel times and capture zones of a site called NWIRP situated in Texas. The objective is to identify locations where monitoring wells should be installed to reduce possible contamination. The method used is a heterogeneous MODFLOW-NWT model to simulate flow and transport together with MODPATH to determine the capture zones, and contains two fractured layers and heterogeneous hydraulic conductivities. They also used a NSMC nonlinear uncertainty analysis technique to generate probability distributions of the calibrated parameters and particle travel-time predictions. In the paper, some issues have to be better explained/corrected:

Comment#1: line 129: Is it a good approximation to relate rock densely fractured to a porous medium?

Response: Modeling the system as an EPM was appropriate for this site. Our other choice would have been to treat the site as a DFN; however, the fracture system has not been characterized for this site (e.g., no data on fracture length, spacing, aperture, orientation, etc.). On the other hand, densely fractured rock significantly enhances hydraulic conductivity of a rock formation and our assumed hydraulic conductivity of the weathered, upper layer is about five orders of magnitude higher than the lower layer. This high hydraulic conductivity supports the assumption of modeling the densely fractured rock as an EPM. The following text was added to the revised manuscript: “The upper layer in the model domain is densely fractured Figure 5; however, the fracture system has not been characterized for this site (e.g.,~no data on fracture length, spacing, aperture, orientation, etc.). A~dense fracture system significantly enhances hydraulic conductivity of a rock formation. Sufficient data were available to characterize the system as an equivalent porous medium with enhanced hydraulic conductivity for the upper layer compared to the lower layer.”  Please see page 5 and lines 142-146. 

Comment#2: line 135: I do not understand how many cells contain the model. Is it 2 (layers) x 126 columns x 97 rows? then, each cell has an area of 100 x 100 m2? This is a huge area for a single cell in the grid and I am not sure that results might be reliable.

Response: We agree with the reviewer that cell size is fairly large. We conducted a grid-refinement study by making the grid size 10×10 m2. This did not improve the simulated hydraulic heads of the site nor significantly change the flow paths; therefore, we kept the grid size as 100×100 m2. Because many thousands of model runs were required for the nonlinear analysis, the large cell sizes made this effort computationally tractable. 

Comment#3: line 127: a constant value of the recharge is not a good approximation for such a complex site.

Response: We agree with the reviewer; a transient model would be best for this site. We discussed this on lines 348-353 in the original manuscript where we noted that we do not have transient water-level data. Therefore, we could not implement a transient recharge rate. However, two base-flow studies (M, S) strongly supported this recharge rate.

Comment#4: line 344: "This study also demonstrated that if a contaminant reached the upper layer (for example during 342 storm events that raise the water table), it will travel much faster to surrounding streams." I do not see how this might be possible if the recharge was set to be a constant value and there is not a water-level time series data (line 349).

Response: This is entirely physically plausible and precisely what happened at this site as far as we can tell. The reviewer is correct that this model does not directly simulate this phenomenon. However, with a saturated upper layer with high hydraulic conductivity and particles released from both the upper and lower layers, we approximated this condition. We regret not clarifying this point in our manuscript. In the reversed manuscript, we have added the following text: “Due to lack of transient recharge and head data, a steady-state numerical model (see Subsection 2.2) was developed assuming that the upper layer was fully saturated during the storm periods and five orders of magnitude more conductive than the bottom layer.”  Please, see page 5 and line 138-140.

Reviewer 2 Report

The authors analyzed uncertainty reduction for groundwater travel time and others using MODFLOW and PEST after building a steady-state flow model for the Naval Weapons Industrial Reserve Plant. I think the topic is very interesting and this work was well written, so I recommend acceptance upon minor revision.

1) Introduction: please add a few sentences to explain the scientific question addressed by this work, such as uncertainty reduction by identifying the well location that intercepts most tracers, so that the paper reads more than a project report.

2) Model explanation: please consider adding a short explanation for:

(a) why the no-flow boundary condition (but not the others such as a free or ghb boundary) was used

(b) why dispersion was not considered for travel time (we know that matrix diffusion and other trapping processes or local/regional mixing can cause a PDF of travel times, which may be broader than the one shown in Fig. 11 caused by parameter uncertainty)

(c) how streams were modeled/validated in your MODFLOW model, and why they look like the only ending points of (backward?) particle tracking.

Author Response

Comments and Suggestions for Authors

The authors analyzed uncertainty reduction for groundwater travel time and others using MODFLOW and PEST after building a steady-state flow model for the Naval Weapons Industrial Reserve Plant. I think the topic is very interesting and this work was well written, so I recommend acceptance upon minor revision.

Comment#1: Introduction: please add a few sentences to explain the scientific question addressed by this work, such as uncertainty reduction by identifying the well location that intercepts most tracers, so that the paper reads more than a project report.

Response: The authors thank the reviewer for this comment. We added the following sentences to the Introduction: “Based on linear and nonlinear uncertainty analyses using MODFLOW and MODPATH models of the NWIRP site, we quantified the times for particles to reach nearby streams, which ultimately feed into nearby rivers and lakes. We also suggested locations for additional monitoring wells and a future data collection plan to better characterize and monitor the site to support improved waste-containment operations. These additional data were selected to ensure the greatest reductions in model-prediction uncertainty.” See page 2 and lines 45-50.

Comment#2: Model explanation: please consider adding a short explanation for:

(a) why the no-flow boundary condition (but not the others such as a free or ghb boundary) was used

Response: Thank you for pointing this out. The model was a simple steady-state model where we tried to simulate hydraulic heads. No-flow BCs helped to make the model simple and this kept runtimes appropriately short for an effort requiring thousands of model runs. However, to mitigate no-flow boundary effects, a 1.5 km buffer zone was added to the edge of the model surrounding the area of interest and this was explained in lines 156-157.  We applied recharge as the specified-flux boundary at the top of the upper layer of the model and head-dependent flux BCs for streams. We added the following text in the updated manuscript to clarify the BCs: “In addition to specified-flux and no-flow BCs, head-dependent flux (HDF) BCs were assigned to cells containing a stream (drain cells in MODFLOW). In the HDF BC, a reference head and a conductance value were assigned at stream boundaries where water may only leave the flow system through the drain cell and it may not re-enter the groundwater system through cells further downstream. Also, if the hydraulic head in the stream cell fell below a certain threshold (e.g., bottom of the upper layer), the flux from the drain to the model cells was set to zero.” See page 5 and lines 157-163.

(b) why dispersion was not considered for travel time (we know that matrix diffusion and other trapping processes or local/regional mixing can cause a PDF of travel times, which may be broader than the one shown in Fig. 11 caused by parameter uncertainty)

Response: We agree. Considering the tracer as chemical components would better characterize transport through the site, but requires site data for dispersion, matrix diffusion, sorption, decay, and other local or regional mixing phenomena, which were not available. Relegated to particles, we used MODPATH to simulate particle travel times as conservative estimates for contaminant transport. We agree that this must be discussed in the manuscript and the following text was added, “Also, chemical wastes were treated as conservative particles. Although a more complex transport model would be of value, the necessary data (e.g., a comprehensive study on the chemical waste including measurements of dispersion, matrix diffusion, sorption, decay, and other local or regional mixing phenomena for individual chemical constituents) are not available.” See page 16 and lines 401-404.

(c) how streams were modeled/validated in your MODFLOW model, and why they look like the only ending points of (backward?) particle tracking.

Response: The authors thank the reviewer for pointing out this potential source of confusion. Streams were treated as drains. Please see Response Minor Comment 8 of Reviewer 3. Please see page 10 and line 323-324.

Reviewer 3 Report

Review Manuscript no. 965471 "Post-calibration Uncertainty Analysis for Travel Times at a Naval Weapons Industrial Reserve Plant" by Ahmmed et al.to Water

General comments

Introduction is missing a thorough discussion of the advantages and disadvantages of the techniques employed in the uncertainty assessment (e.g. NSMC). Also I couldn't find a clear description on the novelty and contribution of this work to specialized knowledge. Without this is difficult to assess the value of this work.

A geological map describing the Georgetown Limestone and geological context is required. Same as input for the conceptual model discussion.

Much of the focus of the article is on the calibration methodology and post-assessment with little emphasis on advancing the specialized knowledge. As such, large portions of section 2 (Modelling Approach) could be moved to Introduction as part of the state-of-the-art discussion.

L336-351. This section as it is written now would fit better the conclusions section.

Major comments

Not enough background information to link pollutant storage and migration. Where is the perchlorate stored? upper highly-fractured or lower moderately fractured? surface? whole aquifer section? If perchlorate is mobilized from lower aquifer to upper aquifer during storms, it means it must have reached the lower aquifer first. How? preferential flowpaths?

The claim water table is sensitive to recharge (storms) (Fig 2.) is not evident nor properly justified. All piezometers in Fig. 2 show a similar signal for the period. They respond similarly to the first recharge pulse but do not respond to subsequent pulses (discharge-recharge cycles). I guess if trend is removed from the piezometers signals this will be more evident.

Section 2.1. As any other modelling exercise a thorough conceptualization of the modeled area is required. This component is a bit weak in this work. This is essentially a local-scale model, how it fits the regional landscape? Is steady-state justifiable? is EPM approach justifiable for upper aquifer considering mobility of pollutant? role of preferential flowpaths if any?

I wonder how NSMC deals with starting conditions for k-fields and the possibility of local minima. My understanding is that once a calibrated k-field is obtained (solution-space) this is perturbed by null-space uncertainty. If this initial solution is around a local minimum there is no way to ensure is the global solution to the objective function (minimization), unless the the solution space is re-started a number of times from different calibrated solutions. I guess this is a discussion missing in the text.

L257-265. I am not sure I understand the thinking behind this analysis. The closer the well to the flowpath of interest the larger the impact. If that's the case then analysis is just function of the location where the particles are released forward in time. You might as well repeat the analysis from the destination to origin using a backward particle tracking and see if analysis remains valid.

L291-296. It is interesting how this statement could easily invalidate the assumption of using a kriging interpolation for k-fields within NSMC. The whole purpose of using kriging is to obtain smooth interpolated k realizations which might fail to capture high degrees of actual heterogeneity. Although you could claim this is captured by the nugget effect in the variogram, only sill and range are discussed. This could be specially true for a fractured aquifer system as this one, which was further conceptualized as EPM. Given the whole purpose is to investigate flowpaths and fate of particles, I believe this is a major issue.

L324-328: This is also an important understatement. Constraining the calibration with hydraulic heads is relatively of low information value and adds little to identifiability of the conceptual model. Calibration should use a combination of heads, fluxes and other data (e.g. travel time) for maximum information value.

Section 4. Conclusions. This is really a summary of results. There is little insight on the techniques employed, implications for water and pollutant management, how to advance the knowledge in the area, recommendations for future research, etc.

Minor comments

Figure 1: source location? lake location for discharge of pollutants?

Figure 4: could you provide some more information for fracture characterization: density, orientation, mean openings, etc.

L142: drain cells in MODFLOW?

Figure 5: is this one k realization or average?

L244-245: this is interesting given that the main issue highlighted in the introduction is the vertical migration of pollutant due to raising heads by storms recharge.

L249: Figure 7: What does it mean (in practice) that mainly horizontal conductivity and anisotropy for the upper layer are the most identifiable parameters ?

L280: what about hu?

L287: this needs to be defined somehow as it conditions performance of the analysis.

L296: where in the streams? please specify + typo in sentence

Recommendation

Based on the above review, I am left to believe reviews are above average. I would suggest the authors the following actions: expand on the introduction (contribution, novelty, state-of-the-art) to provide reviewers with a good basis for a fair assessment of this work, present a detailed conceptual model of the study site, provide discussion on advantages and disadvantages of the techniques employed (claiming it shows a similar performance than MCMC does not seem enough), go beyond the purely technical results in explaining the implications of this work (e.g. new monitoring network, transient processes, vertical heterogeneity, validity of smooth vs heterogeneous k-fields for flowpath analysis, etc.).

Author Response

General comments

Introduction is missing a thorough discussion of the advantages and disadvantages of the techniques employed in the uncertainty assessment (e.g. NSMC). Also I couldn't find a clear description on the novelty and contribution of this work to specialized knowledge. Without this is difficult to assess the value of this work. A geological map describing the Georgetown Limestone and geological context is required. Same as input for the conceptual model discussion. Much of the focus of the article is on the calibration methodology and post-assessment with little emphasis on advancing the specialized knowledge. As such, large portions of section 2 (Modelling Approach) could be moved to Introduction as part of the state-of-the-art discussion.

Response: There is plenty of literature related to Markov Chain Monte Carlo (MCMC) and Null Space Monte Carlo (NSMC) simulations. We provided a short introduction of these because comparing MCMC nor NSMC was not the main goal of this study. However, we cited work that has thorough explanations of these approaches for readers to peruse at their leisure.  In terms of novelty, we agree with the Reviewer. We neither developed any new techniques nor claimed such; rather we demonstrated a valuable approach for thoroughly investigating the site with a straightforward numerical flow and transport model. We applied existing knowledge to show that a simple hydrogeologic model along with post-calibration uncertainty analyses support critical decision making to help avoid human health risks that the site authority probably should have implemented early on in the remediation process. A geological map illustrating Georgetown Limestone might be appropriate, but adds minimal value to our present study and would serve to unnecessarily lengthen the manuscript. Our numerical method captured the heterogeneous geology through pilot-point hydraulic conductivity estimates as explained in Subsection 2.3. We modified the Introduction based on valuable comments from all reviewers. Please see changes to the Introduction.

L336-351. This section as it is written now would fit better the conclusions section.

Response: We agree and this section was moved to the Conclusions. See page 15.

Major comments

Major Comment#1: Not enough background information to link pollutant storage and migration. Where is the perchlorate stored? upper highly-fractured or lower moderately fractured? surface? whole aquifer section? If perchlorate is mobilized from lower aquifer to upper aquifer during storms, it means it must have reached the lower aquifer first. How? preferential flow paths?

Response: We added the following text: “stored several hazardous chemical wastes including ammonium perchlorate at five administrative locations: G, H, L, M, and S (see Figure 1).” Please see page 1 and lines 22-24. 

Major Comment#2: The claim water table is sensitive to recharge (storms) (Fig 2.) is not evident nor properly justified. All piezometers in Fig. 2 show a similar signal for the period. They respond similarly to the first recharge pulse but do not respond to subsequent pulses (discharge-recharge cycles). I guess if trend is removed from the piezometers signals this will be more evident.

Response: We think that the reviewer asked us to remove recharge rate from the measured precipitation. If we removed a static recharge rate of 7% of precipitation reported by Myrick (1989) and Cannata (1988), then the net effect will be unchanged. No changes were made for this comment.

Major Comment#3: Section 2.1. As any other modelling exercise a thorough conceptualization of the modeled area is required. This component is a bit weak in this work. This is essentially a local-scale model, how does it fit the regional landscape? Is steady-state justifiable? Is the EPM approach justifiable for upper aquifers considering the mobility of pollutants? role of preferential flow paths if any?

Response: The modeled section of the aquifer system is representative of the regional aquifer, the Washita Prairie segment of the Edwards (Balcones Fault Zone) Aquifer (Yelderman). Although a transient model might strengthen the research, transient site data were never collected (only a few of the wells had measurements available at different times). Nevertheless, we believe the steady-state model contributes significantly to increased understanding of this aquifer system and that the results can be extrapolated to similar aquifers. Because the fracture network in this section of the aquifer was due more to weathering than faulting in the Washita Prairie segment and because there are no known discrete fracture systems in the area, the EPM approach is reasonable. There are no known preferential flow paths in this domain at the scale of the model and therefore insufficient data to characterize a DFN approach anyway. No changes were implemented for this comment.

Major Comment#4: I wonder how NSMC deals with starting conditions for k-fields and the possibility of local minima. My understanding is that once a calibrated k-field is obtained (solution-space) this is perturbed by null-space uncertainty. If this initial solution is around a local minimum there is no way to ensure it is the global solution to the objective function (minimization), unless the solution space is re-started a number of times from different calibrated solutions. I guess this is a discussion missing in the text.

Response: This is a valid concern and we agree with the reviewer. The NSMC process includes generating random parameter fields using a pre-calibration covariance matrix centered on the calibrated parameter field with subsequent removal of solution-space uncertainty from the random distributions. The differences between the calibrated parameter field and the set of parameter fields generated from the PEST utility RANDPAR and modified by PNULPAR lie in the null space.  Model results (hydraulic heads and particle travel times) using the newly generated parameter fields are necessarily similar to the results from the calibrated model; however, up to three three calibration iterations (each yielding an updated Jacobian matrix and set of calibrated parameters) were undertaken to see if these steps could bring the model back into calibration. If a sufficiently low objective function could not be achieved after these three calibration iterations, then that parameter field was discarded (as not effectively calibrating the model or maintaining parameter reality). We hope that we have responded appropriately to the reviewer’s comment, please see lines 110-112. We did not discuss this in detail because this is outside the scope of this study and also there is a lot of literature on PEST’s Null-space Monte Carlo technique. No changes were made for this comment.

Major Comment#5: L257-265. I am not sure I understand the thinking behind this analysis. The closer the well to the flowpath of interest the larger the impact. If that's the case then analysis is just a function of the location where the particles are released forward in time. You might as well repeat the analysis from the destination to origin using a backward particle tracking and see if analysis remains valid.

Response: It is true that if we were seeking information on the source of contamination that we could run backward particle tracks and then locate wells to most reduce uncertainty. It is not specifically about the well location, but the location of the well with regard to the pilot points. It is intuitive that pilot points located in closer proximity to a well (observation) will be better constrained by those data. No changes were made related to this comment.

Major Comment#6: L291-296. It is interesting how this statement could easily invalidate the assumption of using a kriging interpolation for k-fields within NSMC. The whole purpose of using kriging is to obtain smooth interpolated k realizations which might fail to capture high degrees of actual heterogeneity. Although you could claim this is captured by the nugget effect in the variogram, only sill and range are discussed. This could be especially true for a fractured aquifer system as this one, which was further conceptualized as EPM. Given the whole purpose is to investigate flowpaths and fate of particles, I believe this is a major issue.

Response: We agree. We did not use the nugget effect to capture site heterogeneity. However, our broad hydraulic conductivity constraints during model calibration effectively interrogated a broad uncertainty range and a high degree of heterogeneity. Moreover, when interrogating the semivariogram, there was no obvious need for adding the nuggets effect. There is a lot of subjectivity when assigning a model to the semivariogram data and we have done this to the best of our expert judgment. Note that the second author teaches the Geostatistics class at Baylor and has thoroughly reviewed the approach. No changes were made for this comment. 

Major Comment#7: L324-328: This is also an important understatement. Constraining the calibration with hydraulic heads is relatively of low information value and adds little to identifiability of the conceptual model. Calibration should use a combination of heads, fluxes and other data (e.g. travel time) for maximum information value.

Response: It is known that calibrating a flow model to heads only, without a flow/travel-time measurements, can lead to an under-constrained optimization. Unfortunately, no flow/travel-time measurements were available for us to directly include in the calibration. Fortunately, all of the boundaries were no-flow except for the specified infiltration rate at the water table, which was from site data. Thus, no lateral throughflow was assumed for this site and the boundaries were set sufficiently far from the region of interest. Ultimately, all infiltration flow exited the system at nearby streams, which is a reasonable assumption for the site with its hilly topography and boundaries established away from points of interest. No changes were made for this comment. 

Major Comment#8: Section 4. Conclusions. This is really a summary of results. There is little insight on the techniques employed, implications for water and pollutant management, how to advance the knowledge in the area, recommendations for future research, etc.

Response: We moved the last two paragraphs of the Discussion to the Conclusions and added the following text: “Vertical anisotropy did not play a significant role because there was minimal vertical flow in the model and no well data suggesting a vertical gradient.” See page 15 and lines 372-373. “While a discrete fracture network model could be an alternative approach, the fractured system was not sufficiently characterized to support development of such a model.

Also, chemical wastes were treated as conservative particles. Although a more complex transport model would also be of value, the necessary data (e.g., a comprehensive study on the chemical wastes including measurements of dispersion, matrix diffusion, sorption, decay, and other local or regional mixing phenomena for individual chemical constituents) are not available.” See page 16 and lines 398-404. 

Minor comments

Minor Comment#1: Figure 1: source location? lake location for discharge of pollutants?

Response: Modified. Please see page 1, lines 22-24.

Minor Comment#2: Figure 4: could you provide some more information for fracture characterization: density, orientation, mean openings, etc.

Response: Please see page 5 and lines 142-146. These data would certainly have been nice to have as they could be used to develop a DFN model.

Minor Comment#3: L142: drain cells in MODFLOW?

Response: Yes. Please see page 5 and line 158.

Minor Comment#4: Figure 5: is this one k realization or average?

Response: The cross plot is for the calibrated pilot points and the resulting k distribution. This has been made clear in the manuscript on page 8, line 247.

Minor Comment#5: L244-245: this is interesting given that the main issue highlighted in the introduction is the vertical migration of pollutants due to raising heads by storm recharge.

Response: Storms increase hydraulic heads, but the fracture network governs the flow in the system; therefore, the statement is reasonable and appropriate for this site. No changes were made for this comment.

Minor Comment#6: L249: Figure 7: What does it mean (in practice) that mainly horizontal conductivity and anisotropy for the upper layer are the most identifiable parameters ?

Response: It means flow in the system was constrained by horizontal hydraulic conductivities and horizontal anisotropies. Because none of the wells had packed off screens in both the upper and lower layers, there were no data to constrain this parameter. In our steady-state model, vertical anisotropy had little influence on the flow behavior. No changes were made for this comment.

Minor Comment#7: L280: what about hu?

Response: We added the following text to the updated manuscript: “Among horizontal anisotropies, hu was significantly constrained by the calibration dataset, which reduced hu variance from 0.61 to 0.05 (~92% reduction). However, hl was not constrained by the calibration dataset.” See page 9 and lines 269-271.

Minor Comment#8: L287: this needs to be defined somehow as it conditions performance of the analysis.

Response: The authors do not fully understand this comment. However, if it pertains to the NSMC processes, then the authors refer the reviewer to Subsection 2.5 where we detailed implementation of NSMC.

Minor Comment#8: L296: where in the streams? please specify + typo in sentence

Response: Particles stopped when they reached stream cells. The following text was added to the revised manuscript: “Note that particles may only exit the model through stream cells; therefore,” Please see page 10 and lines 323-324.

Recommendation

Based on the above review, I am left to believe reviews are above average. I would suggest the authors the following actions: expand on the introduction (contribution, novelty, state-of-the-art) to provide reviewers with a good basis for a fair assessment of this work, present a detailed conceptual model of the study site, provide discussion on advantages and disadvantages of the techniques employed (claiming it shows a similar performance than MCMC does not seem enough), go beyond the purely technical results in explaining the implications of this work (e.g., new monitoring network, transient processes, vertical heterogeneity, validity of smooth vs. heterogeneous k fields for flowpath analysis, etc.).

Response: We genuinely appreciate all comments; they have significantly improved the quality of the manuscript. Please see responses to all comments as well as the red text in the revised manuscript.

Bibliography

Myrick, M. “Aquifer-stream interactions and their hydrologic implications in nonkarstic limestones, Washita Prairie, Central Texas: Waco, Texas.” PhD Thesis, Baylor University, 1989, 142 pp.

Cannata, S. “Hydrogeology of a portion of the Washita Prairie Edwards Aquifer, Central Texas: Waco, Texas.” MS Thesis, Baylor University, 1988, 128 pp.

Yelderman, J. "The Washita Prairie segment of the Edwards (Balcones Fault Zone) Aquifer," The Edwards Aquifer: The Past, Present, and Future of a Vital Water Resource, John M. Sharp, Jr., Ronald T. Green, Geary M. Schindel. Geoscienceworld, 20019, https://doi.org/10.1130/2019.1215(10). Accessed 29 October 2020.

Round 2

Reviewer 1 Report

I read the Authors' answer and I believe that this paper has not improved over the previous version. I do not think the fractured zone can be properly modeled by just enhancing the hydraulic conductivity: "A dense fracture system significantly enhances hydraulic conductivity of a rock formation. Sufficient data were available to characterize the system as an equivalent porous medium with enhanced hydraulic conductivity for the upper layer compared to the lower layer."
But the most important point is that the model was simulated using a very small number of cells (2x126x97) and each cell has an area of 100x100 m2 which is a very large area. The Authors say that they used a grid size of 10x10m2 and did not notice significant changes. Without a figure that shows that the results already converge, with respect to the monitoring wells, as for example, it is difficult to understand their analysis and the quality of their choices (hydraulic conductivity, steady-state regime, etc).

Author Response

Comment#2: line 135: I do not understand how many cells contain the model. Is it 2 (layers) x 126 columns x 97 rows? then, each cell has an area of 100 x 100 m2? This is a huge area for a single cell in the grid and I am not sure that results might be reliable.

Response: We agree with the reviewer that cell size is fairly large. However, we conducted a grid-refinement study with four different models. The grid sizes of four models were 100×100 m2, 50×50 m2, 25×25 m2, and 20×20 m2 along x and y directions. (https://github.com/bulbulahmmed/NWIRP). You can verify those models by downloading and decompressing them in Windows. The command to run the models is MODFLOW-NWT.exe followed by the file name with *.nam extension in each directory. Further, we generated the figure below with grid size, time required to run each model, and performances with regard to simulated hydraulic heads. This figure shows that finer grids did not decrease the simulated hydraulic head residuals; therefore, we kept the grid size as 100×100 m2. Because many thousands of model runs were required for the nonlinear analysis, the large cell sizes made this effort computationally tractable. Moreover, we have added the following text and the figure below to the revised manuscript: “A grid-refinement study was performed to adopt an optimal grid resolution for this study. Cell dimensions along x and y axes for the grid-refinement study were 20×20 m2, 25×25 m2, 50×50 m2, and 100×100 m2, each with two layers. Figure 4 shows that finer grids increased run times with negligible accuracy changes for piecewise homogeneous hydraulic conductivities. Therefore, we considered the model with” Please see page 5 and lines 151-155.

Reviewer 3 Report

Thanks to the authors for responding to comments raised. Although I am generally satisfied with the replies/justifications provided in the review I believe explanations provided are not fully reflected in the revised version of the text.

Major comment 2: The main point here is that claiming WT is sensitive to recharge is not properly justified. Only one recharge pulse is observed in piezometers for a short period (Sept 1998), whereas for other pulses no clear relationship is observed (March 1999, May 1999, July 1999). By removing the trend in pp maybe the relationship might be more clear. Otherwise provide a possible explanation for the other pulses not reflected in the water table data.

Major comment 4: Reference to lines 110-112 in the revised text is not a satisfactory reply. A thorough explanation of this issue should be addressed in the text. I disagree with the statement "this is out of the scope of the article" as the main topic of the article is post-calibration uncertainty analysis, and the comments provided in the reply regarding NSMC implementation in PEST should be reflected in the revised version of the article.

Major comment 6: The issue here does not relate to including or not including the nugget effect in the variogram model, but how valid kriging interpolation is for a fractured system approached through EPM. Authors are discussing particle flowpaths and how low-conductivity patches might impact travel times. Low-conductivity patches are a function of representativeness of k-realizations, which are further obtained from kriging interpolations. It is anticipated that these k-realizations might be relatively smooth as an artifact of interpolation thus, underestimating lateral heterogeneity in k. So, this assumption asks for that questions. This has not been addressed nor justified in the text but only in the reply.

Author Response

Major Comment#2: The claim water table is sensitive to recharge (storms) (Fig 2.) is not evident nor properly justified. All piezometers in Fig. 2 show a similar signal for the period. They respond similarly to the first recharge pulse but do not respond to subsequent pulses (discharge-recharge cycles). I guess if trends are removed from the piezometers signals this will be more evident.

Response: First, we would like to apologize for misunderstanding the comment. This comment taught us an important lesson regarding hydraulic heads and their linear trends. Removing linear trends from hydraulic heads makes the figure easier to understand. We removed the previous figure and replaced it with the amended figure below.

Figure 2: Change in head in response to precipitation over time. Here, hydraulic heads are plotted after removing their linear trends.

Major Comment#4: I wonder how NSMC deals with starting conditions for k-fields and the possibility of local minima. My understanding is that once a calibrated k-field is obtained (solution-space) this is perturbed by null-space uncertainty. If this initial solution is around a local minimum there is no way to ensure it is the global solution to the objective function (minimization), unless the solution space is re-started a number of times from different calibrated solutions. I guess this is a discussion missing in the text.

Response: This is a valid concern and we agree with the reviewer. The NSMC process includes generating random parameter fields using a pre-calibration covariance matrix centered on the calibrated parameter field with subsequent removal of solution-space uncertainty from the random distributions. The differences between the calibrated parameter field and the set of parameter fields generated from the PEST utility RANDPAR and modified by PNULPAR lie in the null space.  Model results (hydraulic heads and particle travel times) using the newly generated parameter fields are necessarily similar to the results from the calibrated model; however, up to three three calibration iterations (each yielding an updated Jacobian matrix and set of calibrated parameters) were undertaken to see if these steps could bring the model back into calibration. If a sufficiently low objective function could not be achieved after these three calibration iterations, then that parameter field was discarded (as not effectively calibrating the model or maintaining parameter reality). We hope that we have responded appropriately to the reviewer’s comment, please see lines 110-112. We did not discuss this in detail because this is thoroughly covered in the literature on PEST’s Null-space Monte Carlo technique. We added the following text in the revised manuscript: “Model results (hydraulic heads and particle travel times) using the newly generated parameter fields are necessarily similar to the results from the calibrated model; however, up to three three calibration iterations (each yielding an updated Jacobian matrix and set of calibrated parameters) were undertaken to see if these steps could bring the model back into calibration. If a sufficiently low objective function could not be achieved after these three calibration iterations, then that parameter field was discarded (as not effectively calibrating the model or maintaining parameter reality).” please see page 7-8 and lines 247-253.

Major Comment#6: L291-296. It is interesting how this statement could easily invalidate the assumption of using a kriging interpolation for k-fields within NSMC. The whole purpose of using kriging is to obtain smooth interpolated k realizations which might fail to capture high degrees of actual heterogeneity. Although you could claim this is captured by the nugget effect in the variogram, only sill and range are discussed. This could be especially true for a fractured aquifer system as this one, which was further conceptualized as EPM. Given the whole purpose is to investigate flowpaths and fate of particles, I believe this is a major issue.

Response: We agree. We did not use the nugget effect to capture site heterogeneity. However, our broad hydraulic conductivity constraints during model calibration effectively interrogated a broad uncertainty range and a high degree of heterogeneity. Moreover, when interrogating the semivariogram, there was no obvious need for adding the nuggets effect. There is a lot of subjectivity when assigning a model to the semivariogram data and we implemented our best expert judgment. Note that the second author teaches the Geostatistics class at Baylor and has thoroughly reviewed the approach.  We added the following text in the revised manuscript: “Note, although they admit higher levels of heterogeneity, no nugget effect was considered in this study because the empirical semi-variogram did not require it. Nevertheless, our broad hydraulic conductivity constraints during model calibration effectively interrogated a broad uncertainty range and admitted a high degree of heterogeneity.” Please see page 6 and lines 193-196.

Round 3

Reviewer 1 Report

The Authors have an answer to my question:

"We agree with the reviewer that cell size is fairly large. However, we conducted a grid-refinement study with four different models. The grid sizes of four models were 100×100 m2, 50×50 m2, 25×25 m2, and 20×20 m2 along x and y directions."

Then in the text, the authors wrote:

"A grid-refinement study was performed to adopt an optimal grid resolution for this study. Cell dimensions along x and y axes for the grid-refinement study were 20 × 20 m2, 25 × 25 m2, 50 × 50 m2, and 100 × 100 m2, each with two layers. "

The authors did not adopt the optimal grid resolution that, in this case, would be the one corresponding to 20 x 20 m2.

And the reason is:

"Because many thousands of model runs were required for the nonlinear analysis, the large cell sizes made this effort computationally tractable. "

But I do not think this is a good reason.

Then they wrote:
"Figure 4 shows that finer grids increased run times with negligible accuracy changes for
piecewise homogeneous hydraulic conductivities. "

The sq. difference of head mentioned in Fig 4 indicates RMSE? which is around 100 (units?). In any case, the value is large, and it is not enough to guarantee that it is "good enough."

Author Response

The authors thank the reviewer for diligently reviewing this manuscript. We have attached the response in a pdf file. The authors request the reviewer to look at the response. 

Round 4

Reviewer 1 Report

The paper is now ready to be published on Water.